



# Fluvial Flood Inundation and Humanitarian Impact Model Based On Open Data

Lukas Riedel[1,2], Thomas Röösli[2], Thomas Vogt[3], and David N. Bresch[1,2]

[1]Institute for Environmental Decisions, ETH Zürich, Zürich, Switzerland
[2]Federal Office of Meteorology and Climatology MeteoSwiss, Zürich-Airport, Switzerland
[3]Potsdam Institute for Climate Impact Research (PIK), Potsdam, Germany

**Correspondence:** Lukas Riedel (lukas.riedel@usys.ethz.ch)

**Abstract.** Fluvial floods are destructive hazards that affect millions of people worldwide each year. Forecasting flood events and their potential impacts therefore is crucial for disaster preparation and mitigation. Modeling flood inundation based on extreme value analysis of river discharges is an alternative to physical models of flood dynamics, which are computationally expensive. We present the implementation of a globally applicable, open-source fluvial flood model within a state-of-the-art natural catastrophe modeling framework. It uses openly available data to rapidly compute flood inundation footprints of historic and forecasted events for the estimation of associated impacts. At the example of Pakistan, we use this flood model to compute flood depths and extents, and employ it to estimate population displacement due to floods. Comparing flood extents to satellite data reveals that incorporating estimated flood protection standards does not necessarily improve the flood footprint computed by the model. We further show that, after calibrating the vulnerability of the impact model to a single event, the estimated displacement caused by past floods is in good agreement with disaster reports. Finally, we demonstrate that this calibrated model is suited for probabilistic impact-based forecasting.

## 1 Introduction

Floods are natural hydrological events with significant humanitarian impacts, causing devastation to communities and ecosystems worldwide. They affect millions of people annually, leading to loss of life, displacement, and extensive damage to infrastructure and livelihoods (CRED, 2023). Flood-induced humanitarian crises can strain emergency response systems, with challenges in providing adequate shelter, clean water, and healthcare to affected populations. Over the last years, the number of people exposed to overlapping and compounding natural disasters increased, thus intensifying crises worldwide (IDMC, 2023a). Climate projections indicate that flood frequencies will increase in many regions of the world, as will the overall number of people exposed to floods, in a heating climate (Hirabayashi et al., 2013). Adaptation and mitigation measures are crucial to reducing the humanitarian impacts of floods, and improved forecasting and early warning systems play a significant role in minimizing loss of life.

The UN-adopted Sendai Framework for Disaster Risk Reduction argues for a better understanding of disaster risk to increase resilience (UNISDR, 2015). To that end, decision makers and humanitarian actors require accurate risk assessments and forecasts of imminent events. Impact estimates provide crucial information in disaster preparation and mitigation. Apart



from fatalities, displacement is a particularly severe consequence of disasters and crises. Displacement aggravates local and global food insecurity, and displaced children in particular are at risk of exploitation and abuse, and exposed to malnutrition and disease (UNICEF, 2023). The year 2022 saw a tragically high number of displacements, with more than 32 million internal displacements due to physical hazards alone (IDMC, 2023a). A clear majority of them was caused by floods.

Fluvial floods, pluvial floods, and storm surges show a complex interplay and merging them in a single, comprehensive
modeling approach is an ongoing effort (see Loveland et al., 2021; Eilander et al., 2023). Physical models for exclusively fluvial floods already require an elaborate model cascade (Winsemius et al., 2013). Global climate models (GCMs) provide the meteorological boundary conditions to run hydrological models like LISFLOOD (Van Der Knijff et al., 2010), CaMa-Flood (Yamazaki et al., 2011), or GLOFRIS (Ward et al., 2013). These simulate hydrological processes of catchments, including surface and subsurface runoff, and river routing. For forecasts on shorter timescales and reanalyses, the forcing by GCMs
can be replaced by forecasts and meteorological observations, respectively. The discharge computed by hydrological models serves as the input to high-resolution inundation models for flood dynamics. There are standalone models like CA2D (Dottori and Todini, 2011), but the aforementioned hydrological models all feature fluvial inundation model extensions. Alfieri et al. (2023) recently brought forward an operational flood forecasting and early warning system for the Greater Horn of Africa, demonstrating that flood forecasts based on a full hydrological modeling chain are feasible. However, these modeling chains
are also computationally demanding, and thus may not be suitable for every application. For instance, Sampson et al. (2015) reported that their model required a server cluster with 200 cores to compute flood maps for a $10° \times 10°$ tile at 90 meter resolution in under 24 hours.

Extreme value analysis of river discharge is an established tool to relate predicted and past events. Hirabayashi et al. (2013) first associated return periods to retrospective land-surface model runs to estimate flooded areas and inundations from extreme
river discharges predicted by GCMs. But the historical time series of GCMs and of river discharge reanalysis datasets is limited, and fitted extreme value distributions become uncertain for extreme events with large return periods. Willner et al. (2018) reduced this uncertainty by introducing another extreme value distribution fitted on a pre-industrial GCM run. Still, both studies applied downscaling on the resulting low-resolution flood depth to derive a high-resolution flood fraction. The flood depth information was omitted from the subsequent analyses of affected population. Sauer et al. (2021) and Kam et al. (2021)
retained inundation information in the same approach and used it to estimate economic damages and population displacement, respectively, caused by floods in future climate projections.

In this paper, we present a globally applicable model for rapid mapping of flood inundation footprints, and for computing associated impacts. Instead of employing our own hydrological model, we rely on river discharge data computed by the Global Flood Awareness System (GloFAS; Alfieri et al., 2013), and use global river flood hazard maps by Dottori et al. (2016b) to
estimate flood depths based on local extreme value analysis. The flood model can be applied on river discharge (ensemble) forecasts and reanalysis alike, and computes flood inundation maps for entire countries in few minutes. It is implemented as a Python module of the natural catastrophe impact model CLIMADA, which serves as platform for both climate risk assessment (Aznar-Siguan and Bresch, 2019) and impact-based forecasts (Röösli et al., 2021). We further employ the flood model in



CLIMADA for estimating population displacement due to river floods in Pakistan. We demonstrate that it can be calibrated to reported displacement data, and thus used in comparative event studies, event detection, and impact-based forecasting.

## 2    Data

The GloFAS provides global data on river discharge (Alfieri et al., 2013; Harrigan et al., 2023). Its hydrological modeling chain uses the LISFLOOD hydrological model developed at the Joint Research Centre (JRC), and meteorological forecast data from the European Centre for Medium-Range Weather Forecasts (ECMWF), among many other data sources. Version 3 of the GloFAS model has a time step of 24 hours and outputs the 24 hour mean river discharge on a $0.1°$ grid. The data provided is especially suitable for our task because a historical reanalysis dataset with discharge data starting from 1979 is published among daily ensemble forecasts (Harrigan et al., 2020). Since the same model is used to compute all GloFAS products, forecasted and historical time series can be compared without the need for model error or bias correction. Daily GloFAS forecast and reanalysis data is uploaded to the Copernicus Climate Data Store (C3S, 2023a, b) and can be downloaded via a web interface and a Python application programming interface (API).

Flood hazard maps display the flood inundation and extent for river systems assuming a flood with a specific return period. Dottori et al. (2016b) used GloFAS reanalysis data from 1979 to 2015 to develop global flood hazard maps of river systems with catchment areas greater than 5,000 $km^2$ for flood return periods of 10 yr, 20 yr, 50 yr, 100 yr, 200 yr, and 500 yr, at a resolution of $30''$ (arc seconds). These maps are freely available from the JRC Data Catalogue (Dottori et al., 2016a). Notably, the models of Dottori et al. (2016b) do not include information on flood defenses and control mechanisms, apart from large-scale physical structures that are incorporated in the digital elevation model (DEM).

One particular effort to provide globally consistent flood protection information is the global database on flood protection standards (FLOPROS; Scussolini et al., 2016). This database merges empirical sources, policy specifications, and data from a model relating per capita wealth with flood protection on the sub-national level. However, information on control structures like dams and reservoirs, and especially their management, are not considered. Among the information layers provided by the database, we exclusively select the "merged layer" as the best guess for flood protection standards from the available information.

## 3    Flood Model

To compute a flood inundation footprint from gridded, geo-located river discharge data, said data is related to the historical discharge time series via an extreme value analysis, and the corresponding return period is used to look up flood depths in flood hazard maps. In a pre-processing step, the historical time series of discharges is analyzed for each grid point by fitting a Gumbel distribution. For any discharge input data, the algorithm then computes a flood footprint by

- computing the return period for the input data using the locally fitted Gumbel distributions,

- regridding the return period data onto the grid of the flood hazard maps using bilinear interpolation,



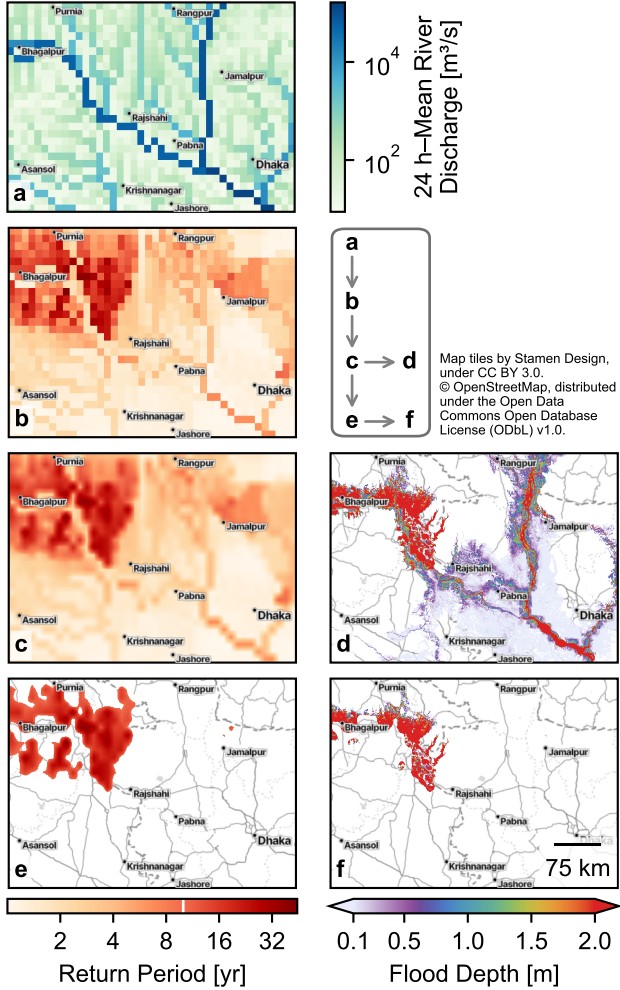

**Figure 1.** Exemplary data transformation within the flood inundation model. The map covers the confluence of the rivers Ganges/Padma and Brahmaputra, and its upstream area in Bangladesh and India. **a**: 24 h–mean river discharge from the reanalysis dataset of GloFAS for 2 August 2007, during a period of floods along the Ganges/Padma river. **b**: Return period computed from the river discharge data. **c**: Return period regridded onto the flood hazard map grid using bilinear interpolation. **d**: Inundation footprint resulting from interpolation of flood hazard maps at each position based on the return period. **e**: Return period after applying estimated flood protection standards, which ignores any return period below the protection threshold. For demonstration, we chose a threshold of $r = 10\,\mathrm{yr}$ (indicated by the white line in the lower left colorbar) for the entire domain. **f**: Inundation footprint from **e**. Note the logarithmic scales for discharge and return period.

90    – optionally applying the FLOPROS protection standard, and

    – deriving a flood depth at every location by interpolating the respective depths in the flood hazard maps using the computed return period.





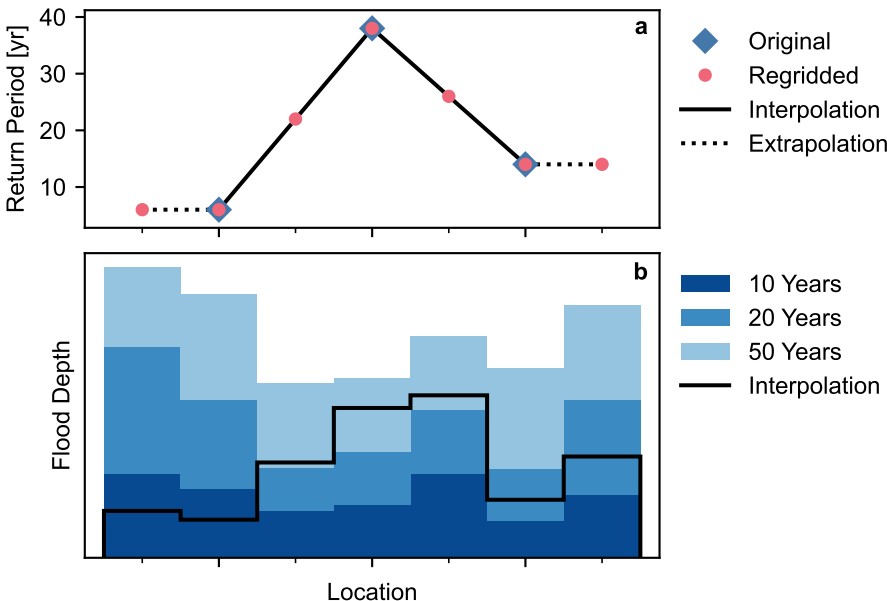

**Figure 2.** Visualization of interpolation for computing local flood depth in one spatial dimension ("Location"). **a**: During regridding, return periods are interpolated linearly (solid line) from the coarser grid (blue markers) to the finer grid (red markers). For points outside the coarser grid, nearest-neighbor extrapolation is applied (dotted line). **b**: At each location, the values of the flood maps for the given return periods (blue shades) are interpolated using the regridded return period values, yielding the output flood footprint (black line).

Fig. 1 visualizes exemplary data at each step of the algorithm. In the following subsections, these steps are explained in detail.

### 3.1 Time Series Analysis

In this pre-processing step, an extreme value analysis is applied to the historical discharge data. To that end, a right-handed Gumbel distribution is fitted to the yearly maximum of the discharge time series at every location independently using the "scipy" Python package (Virtanen et al., 2020).

The right-handed Gumbel distribution is defined by the probability density function (PDF)

$$p_Q(q; \mu, \beta) = \beta^{-1} \exp\left(-\left[z + \exp(-z)\right]\right),$$
$$\text{with } z = (q - \mu)/\beta, \tag{1}$$

where $q$ is a realization of the random variable $Q$ (here: the river discharge), and $\mu$ and $\beta$ are location and scale parameters, respectively. Dankers and Feyen (2008) point out that for calculating return periods from river discharge, a three-parameter generalized extreme value distribution does not yield a clear improvement over the two-parameter Gumbel distribution if the time series is short.



For fitting the two parameters $\mu$ and $\beta$, we employ the "method of moments" (cf. Dottori et al., 2016b), which minimizes the L2-error between the first two moments (mean and variance) of the data distribution and the fitted Gumbel distribution,

$$\text{L2} = \left[\frac{E[\hat{Q}] - E[Q]}{E[\hat{Q}]}\right]^2 + \left[\frac{\text{Var}(\hat{Q}) - \text{Var}(Q)}{\text{Var}(\hat{Q})}\right]^2, \tag{2}$$

where $E$ is the expected value, Var is the variance, $\hat{Q}$ is the discharge data used for fitting, and $Q \sim p_Q$ is the random variable defined by the fitted Gumbel distribution with PDF given by eq. (1).

This process yields a pair of parameters $(\mu_{\boldsymbol{x}}, \beta_{\boldsymbol{x}})$ for every location $\boldsymbol{x}$ on the grid covered by the GloFAS river discharge datasets. The number of samples for fitting each distribution is the number of years included in the reanalysis, up to the date of the publication by Dottori et al. (2016b), $N_f = 36$. The fitted parameters are then stored to avoid repeating this procedure every time a flood footprint has to be computed.

## 3.2 Return Period Computation

After the previous pre-processing step, the model computes an equivalent return period for discharge input data $q_{\boldsymbol{x}}$ at every location $\boldsymbol{x}$. The cumulative distribution function of the fitted Gumbel distribution,

$$F_{Q,\boldsymbol{x}}(q_{\boldsymbol{x}}) = \int_{-\infty}^{q_{\boldsymbol{x}}} p_Q(t; \mu_{\boldsymbol{x}}, \beta_{\boldsymbol{x}}) \, \mathrm{d}t, \tag{3}$$

gives the probability of a discharge less or equal to $q_{\boldsymbol{x}}$ occurring at location $\boldsymbol{x}$ within a year. We interpret the complementary probability as the exceedance frequency

$$f_{\boldsymbol{x}}(q_{\boldsymbol{x}}) = [1 - F_{Q,\boldsymbol{x}}(q_{\boldsymbol{x}})] \, \text{yr}^{-1}. \tag{4}$$

The inverse of that frequency is the return period of the event that the yearly maximum of discharges exceeds $q_{\boldsymbol{x}}$,

$$r_{\boldsymbol{x}}(q_{\boldsymbol{x}}) = [f_{\boldsymbol{x}}(q_{\boldsymbol{x}})]^{-1} \quad \in [1\,\text{yr}, \infty\,\text{yr}). \tag{5}$$

The return period computation transforms the data visualized in Fig. 1a to that in Fig. 1b.

The historical time series only spans 36 years, which results in strongly increasing uncertainty for discharges $q$ with $r(q) > 36\,\text{yr}$. We employ parametric bootstrap sampling to represent this uncertainty (Kyselý, 2008). In this approach, new extreme value distributions are created by drawing samples from the fitted one, and these new distributions are then used to compute a set of return periods using the aforementioned equations. More specifically, $N_f$ samples are drawn from the Gumbel distribution, where $N_f$ is the number of data points used to fit the original distribution as defined above. These samples are used as data points to fit a new Gumbel distribution, and that distribution is then inserted into eqs. (3) to (5). The process is repeated $N_s$ times, yielding an ensemble of return periods

$$r_{\boldsymbol{x}}^{(i)}(q) = \left[1 - F_{Q,\boldsymbol{x}}^{(i)}(q)\right]^{-1} \text{yr}, \quad 1 \leq i \leq N_s, \tag{6}$$



which represents the uncertainty in the return period computation. The sampling density $N_s$ can be chosen by the user. A higher density more accurately represents the uncertainty in the return period computation, but also escalates the computational cost of the subsequent steps.

### 3.3 Geospatial Regridding

The spatial resolution of the GloFAS discharge data ($0.1°$ as of version 3) and the flood hazard maps ($30''$) by Dottori et al. (2016b) differs significantly. We therefore regrid the return period data onto the grid of the flood hazard maps using the geospatial "xESMF" tool with bilinear interpolation (Zhuang et al., 2023).

  Because the GloFAS data is coarser, there are locations near coastlines where flood hazard map data points lie "outside" the GloFAS data grid. To cover these locations, we employ a nearest-neighbor extrapolation of the return period data. The
regridding is visualized in Fig. 2a, and transforms the data displayed in Fig. 1b to that in Fig. 1c.

### 3.4 Flood Protection Standards

The FLOPROS database contains data on return periods associated with modeled flood protection standards (Scussolini et al., 2016). We consider the effect of protection measures by setting the return period $r_{\boldsymbol{x}'}$ to zero if it is lower than protection standard $r_{\boldsymbol{x}'}^{\text{FLOPROS}}$ at the same location $\boldsymbol{x}'$,

$$
r_{\boldsymbol{x}'} = \begin{cases} r_{\boldsymbol{x}'} & \text{if } r_{\boldsymbol{x}'} \geq r_{\boldsymbol{x}'}^{\text{FLOPROS}}, \\ 0\,\text{yr} & \text{else.} \end{cases}
\tag{7}
$$

As zero is an invalid return period according to eq. (5), the resulting values at the affected locations are effectively discarded. This step is optional, and users may choose between different FLOPROS information layers. The application of flood protection standards transforms the data visualized in Fig. 1c to that in Fig. 1e.

### 3.5 Flood Footprint

Finally, the flood footprint related to the discharge input data is created by interpolating a flood depth value from the flood hazard maps. The flood hazard maps define a scalar field $z_{\boldsymbol{x}'r}$ with three-dimensional coordinates: the location $\boldsymbol{x}'$ and the return period $r$. The value of said field is the flood depth at the specified coordinates. To receive the flood depth for a particular return period $r_{\boldsymbol{x}'}$ at location $\boldsymbol{x}'$, we interpolate the field at this location linearly in the return period dimension,

$$
\tilde{z}_{\boldsymbol{x}'}(r_{\boldsymbol{x}'}) = z_{\boldsymbol{x}'r^-} \times \frac{r^+ - r_{\boldsymbol{x}'}}{r^+ - r^-} + z_{\boldsymbol{x}'r^+} \times \frac{r_{\boldsymbol{x}'} - r^-}{r^+ - r^-},
\tag{8}
$$

where $r^{\pm}$ indicates the lesser and greater return periods closest to $r_{\boldsymbol{x}'}$, and $z_{\boldsymbol{x}'r^{\pm}}$ are the associated inundation values in the flood hazard maps. Additionally, we assume that for the lowest return period $r = 1\,\text{yr}$, the flood depth is always zero, and that the maximum flood depth cannot exceed the inundation associated with the maximum return period for which hazard maps are





available, $r_{\max} = 500\,\mathrm{yr}$. With this, we define the flood depth as

$$
z_{\boldsymbol{x}'}(r_{\boldsymbol{x}'}) = \begin{cases} 0\,\mathrm{m} & \text{if } r \leq 1\,\mathrm{yr}, \\ \tilde{z}_{\boldsymbol{x}'}(r_{\boldsymbol{x}'}) & \text{cf. eq. (8), if } 1\,\mathrm{yr} < r_{\boldsymbol{x}'} < r_{\max}, \\ z_{\boldsymbol{x}' r_{\max}} & \text{else.} \end{cases} \tag{9}
$$

The flood depth interpolation is visualized in Fig. 2b. The flood footprint computation transforms the data displayed in Fig. 1c to that in Fig. 1d, and the data in Fig. 1e to that in Fig. 1f.

## 4 Implementation

CLIMADA is a natural catastrophe impact model that computes risk following the definition by the Intergovernmental Panel on Climate Change (IPCC), whereupon natural risks emerge from the exposure of goods or people to weather- or climate-related

hazards, and their vulnerability towards these hazards (Aznar-Siguan and Bresch, 2019). CLIMADA represents hazards, exposures, and vulnerability in a spatially explicit manner. Within the framework, vulnerability is modeled as an impact function, which takes the local hazard intensity as argument and yields a damage factor. Multiplying this factor with the local exposure returns the local impact.

The presented flood model is implemented as a Python module of CLIMADA (Aznar-Siguan et al., 2023). The code is cur-

rently under review for inclusion in the CLIMADA Petals module (CLIMADA Contributors, 2024). It produces one or multiple `Hazard` objects (the CLIMADA data structure of a geophysical hazard) from the discharge input data, which are downloaded automatically from the Copernicus Climate Data Store (CDS) by employing the "cdsapi" Python package (ECMWF, 2023a) that accesses the CDS API. For accessing, transforming, and storing the multi-dimensional data within our module, we use the "xarray" Python package (Hoyer and Hamman, 2017).

Most of the module functionality is wrapped in the `RiverFloodInundation` class that performs these tasks with minimal user input. Users have to state which GloFAS discharge data to download and can then compute a CLIMADA hazard from this data using default settings. Alternatively, they can add custom settings (like bootstrap sampling) or adjust each step of the model pipeline (Sec. 3) individually. A setup function has to be executed once, which prepares the static data required for all computations. This function downloads the historical discharge data, performs the time series analysis, and stores the

fitted Gumbel parameter distribution data on the user's device. It also downloads the flood hazard maps (given as GeoTIFF files) and merges them into a NetCDF file for easier access. Some of the tasks performed by the module benefit from parallel execution on multiple processors. Where applicable, users may add information on the number of processors a task should use and specify the amount of memory available. In all other cases, the computation pipeline uses the default xarray multithreading to maximize performance.



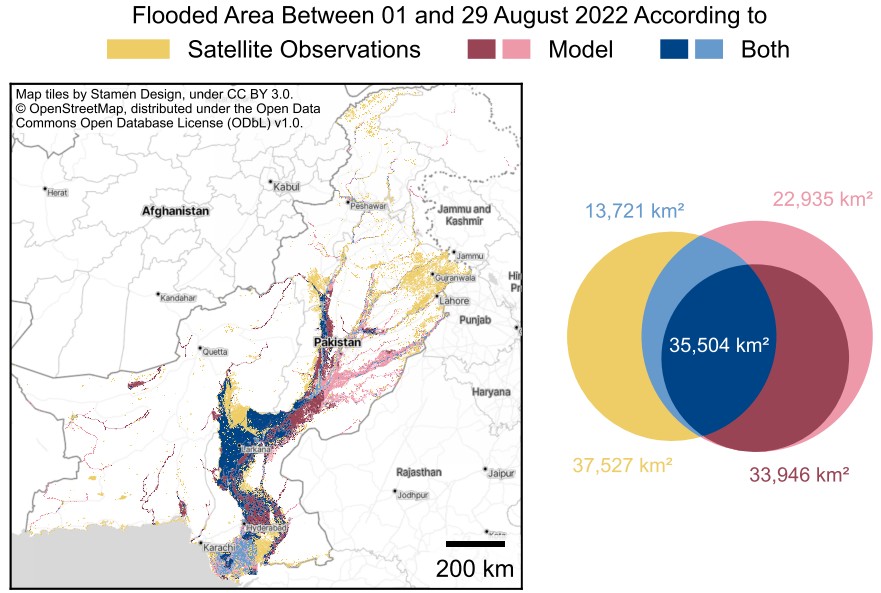

**Figure 3.** Comparison between flood extents in Pakistan computed by the model and observed by VIIRS instruments aboard the NOAA satellites (UNOSAT, 2022). Blue areas denote agreement between observation and flood model. Red areas indicate where the model computed a flood, but no flood was observed. Yellow areas denote where the model did not compute a flood, but floods were observed. Darker and lighter hues of red and blue indicate the model output when considering flood protection standards according to the FLOPROS database, and when not, respectively. **Left**: Map of flood extents in Pakistan. **Right**: Venn diagram of the same data with respective total areas computed in cylindrical equal area projection (ESRI:54034).

## 5    Application to Floods in Pakistan

Pakistan is a flood-prone country and particularly vulnerable to the effects of climate change (Eckstein et al., 2021). In summer 2022, Pakistan experienced its arguably most devastating floods to date. They were caused by the compounding effects of a dry winter season, strong glacier and snow melt due to unusually high temperatures, and a monsoon anomaly with heavy rainfalls that lasted for weeks. The floods affected over 30 million people and left more than 20 million people in need of humanitarian assistance (OCHA, 2022b). Also due to its magnitude, the event gained much international interest and coverage in media and research alike. The variety of data available enables us to calibrate our model to this event. We then put the event in historic perspective and demonstrate its capability of impact-based forecasting for a flood event in 2023.

This section describes the methodology of the model application and its results. The latter will be discussed in Sec. 6.

### 5.1    Flood Extent

In a first step, we computed a flood footprint with our model and compared its spatial extent with satellite observation data. The Humanitarian Data Exchange (HDX) hosts a dataset published by the United Nations Satellite Centre (UNOSAT), which



**Table 1.** Classification metrics for flood extents in Pakistan between 01 and 29 August 2022 computed with the model considering no flood protection, and the FLOPROS protection standards, respectively. Satellite data by UNOSAT (2022) is considered the ground truth against which the model data are compared. See Fig. 3 for a visualization. For each score, value indicating better performance is highlighted. See Appendix Sec. A for the definition of these metrics.

|  | No Protection | FLOPROS |
|---|---|---|
| Precision $P$ | 0.46 | **0.51** |
| Recall $R$ | **0.57** | 0.41 |
| Specificity $S$ | 0.42 | **0.65** |
| F1-score $F_1$ | **0.51** | 0.45 |
| Critical Success Index (CSI) | **0.34** | 0.29 |
| Matthews Correlation Coefficient (MCC) | $-0.01$ | **0.07** |

is based on observations of the Visible Infrared Imaging Radiometer Suite (VIIRS) instruments aboard the NOAA-20 satellites (UNOSAT, 2022, glide number: FL20220808PAK). This dataset contains satellite detected water extents between 01 and 29 August 2022 in Pakistan, the time period of peak flood extents.

To derive an equivalent flood extent from our model, we downloaded the reanalysis river discharge for Pakistan between 01 and 29 August 2022, and selected the maximum discharge at each location. We then employed the flood model to compute a flood footprint. Since the satellite data does not estimate flood depth, we ignored the flood depth computed by the model and simply considered a location flooded if its flood inundation was greater than 10 cm. We considered two separate model runs; one which applied flood protection standards as listed in the FLOPROS database, and one which did not account for

any flood protection. To compare the model output and the satellite observation data, we coarsened the satellite data onto the $30''$-resolution of the model. This process omitted some small-scale features which could not be resolved by our model, but changed the total flooded area of the VIIRS dataset only by about 1%.

 Figure 3 displays the flood extent as calculated by our model for Pakistan between 01 and 29 August 2022, and compares it to the satellite-based observations. Model agreement is good especially in the area around the city of Larkana. East of Hyderabad,

the flood model predicts extensive flooded regions where the satellites only detected patchy flooding. The flood model fails to capture observed floods south-east of Hyderabad, north of Larkana, and around Gujranwala. Differences between the model without flood protection and with FLOPROS protection standards considered are pronounced near the Indus river delta south-east of Karachi, where the "FLOPROS" model fails to predict extensive flooded areas, and further upstream in eastern Pakistan, where the "No Protection" model predicts extensive flooding while little was observed.

To evaluate the accuracy of our model against the observed flood extent, we computed metrics for binary classification. Their definition can be found in Appendix Sec. A. The metrics for the Pakistan floods in August 2022 are displayed in Table 1. When applying FLOPROS flood protection standards to our model, the precision or positive predictive value $P$ increases





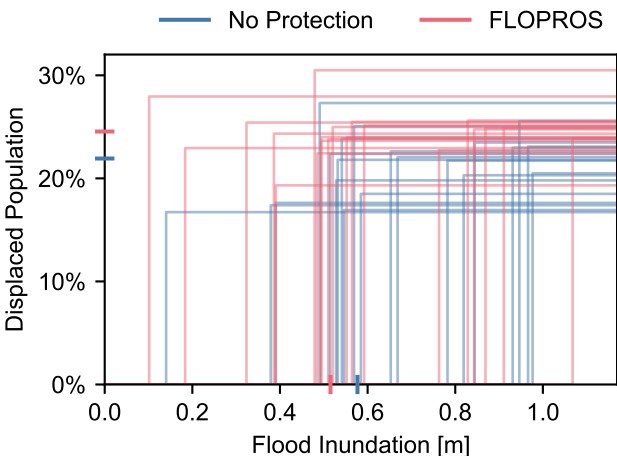

**Figure 4.** Impact functions calibrated with cross-calibration, using data on population displacement in Sindh provided by PDMA (2022). Each displayed function is the result of a single calibration. The colored ticks on the x- and y-axes denote the median of the respective parameter distribution.

from 0.46 to 0.51, and the specificity or true negative rate $S$ increases from 0.42 to 0.65, compared to the model without flood protection standards. However, the recall or true positive rate $R$ decreases from 0.57 to 0.41. This is reflected in the F1-score, which assesses the "No Protection" model performance slightly higher than that of the "FLOPROS" model, with 0.51 against 0.45. Similarly, the critical success index scores 0.34 for "No Protection" against 0.29 for "FLOPROS". The Matthews Correlation Coefficient (MCC) is 0.07 for the "FLOPROS" model, indicating only slightly better prediction than random values. Nonetheless, the MCC for "No Protection" is clearly worse with a value of $-0.01$.

## 5.2 Population Displacement

In this section, we use data on displacement in the Sindh province during the 2022 floods to calibrate the impact functions in our model. The Provincial Disaster Management Authority (PDMA) releases regular situation reports on floods and other disasters, and their impacts, in Sindh. According to these reports, the maximum number of about 7.3 million displaced people was reached by 30 September 2022 (PDMA, 2022). The the PDMA report for this day lists displaced population per each district of the Sindh province except Karachi, where a single number is reported for all six districts. We assumed that this number of displaced people accumulated due to the floods occurring since July 2022, and neglected that some people might already have returned to their homes during that period. We thus chose the maximum discharge from the GloFAS reanalysis data at every point within Pakistan between 01 July and 30 September 2022 as input data for computing a flood footprint. This inundation footprint served as basis for the following impact calculations. Again, we computed one footprint without flood protection and one where the FLOPROS protection level was considered. We selected the latest WorldPop population dataset from 2020 with a resolution of $30''$, or around 1 km at the equator, as exposure (WorldPop, 2020).




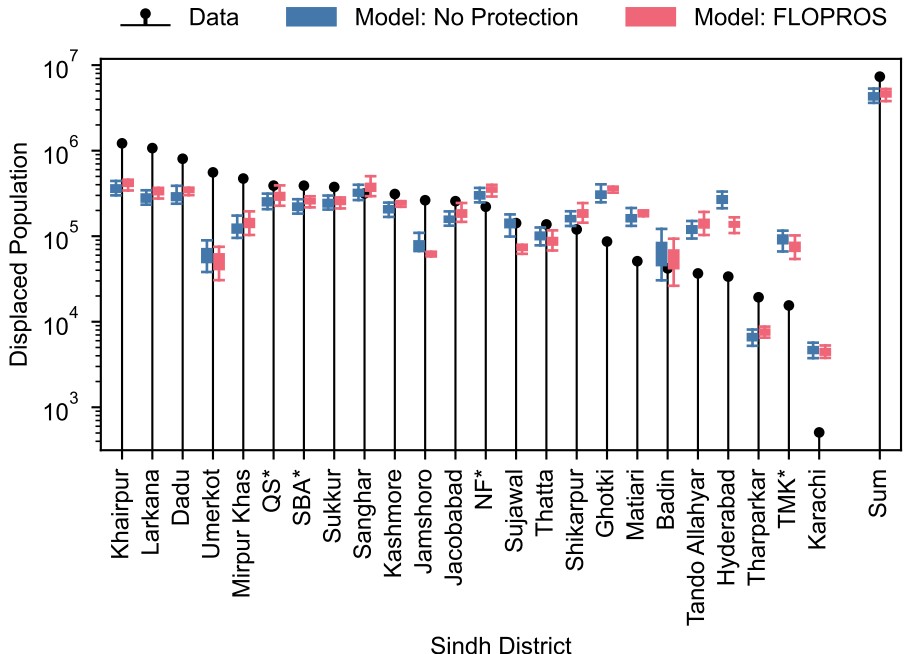

**Figure 5.** Comparison between displacement data provided by PDMA (2022) and the calibrated impact model output for flood footprints without considering flood protection and with FLOPROS protection levels incorporated. The impact distribution for 20 cross-calibrated impact functions each is represented as boxplot; boxes indicate the interquartile range, whiskers delimit the distribution confidence interval, and outliers are omitted. The vulnerability is the only source of uncertainty considered in the impact. "Sum" refers to the sum of all districts. "Karachi" refers to all districts within the Karachi division. *QS: Qambar Shahdadkot, SBA: Shaheed Benazir Abad, TMK: Tando Muhammad Khan, NF: Naushahro Feroze.

We calibrated the impact function parameters using the Bayesian optimization package "BayesianOptimization" (Nogueira, 2014). As generalized impact function, we chose a simple step function with two parameters; the inundation threshold above which displacements occurs, and the percentage of population displaced once displacement occurs. The target function to be maximized was the negative mean squared log error (MSLE) between the PDMA report data and the modeled impact,

$$t(\hat{\boldsymbol{\chi}}) = -\mathrm{MSLE}(\hat{\boldsymbol{\chi}}) = -\frac{1}{N_D} \sum_i^{N_D} [\ln(\chi_i + 1) - \ln(\hat{\chi}_i + 1)]^2, \tag{10}$$

where $\chi_i$ and $\hat{\chi}_i$ are the modeled and the reported number of displaced people for district $i$, respectively, and $N_D$ is the number of districts. This error measure is suitable for data points with varying orders of magnitude, and puts a larger penalty on underestimation than on overestimation. The modeled values $\chi_i$ were calculated by computing the impact for all exposure points in the Sindh province individually, and then summing up the impacts for each district $i$.

Because the dataset is small, calibrating to the entire data will likely overfit the model to this particular flood event. We therefore employed cross-calibration, calibrating the impact function multiple times on data subsets. The subsets were chosen





to contain 20 randomly selected data points from the 24 available districts, ignoring the data and calculated impact for the remaining four districts. We repeated this process 20 times, resulting in a set of 20 calibrated impact functions both for the model without protection measures ("No Protection") and for the model with FLOPROS protection standards considered ("FLOPROS"). These impact functions are visualized in Fig. 4. For the "No Protection" model, the impact threshold ranges from 0.14 m to 0.98 m, with a median of 0.58 m, and the percentage of displaced population from 16.7 % to 27.3 %, with a median of 21.9 %. For the "FLOPROS" model, on the other hand, the threshold ranges from 0.10 m to 1.07 m, with a median of 0.52 m, and the percentage of displaced population from 19.3 % to 30.5 %, with a median of 24.5 %.

The impact estimation for all districts with the varying impact functions is visualized in Fig. 5. For the majority of districts, the difference between median estimated impact and the data is lower than one order of magnitude. However, in most cases, the variation in impact due to the different impact functions is lower than the deviation between median impact estimate and data, meaning that the data is rarely covered by the range of model outputs. Also, the intra-model variance is typically higher than the variation between the "No Protection" and "FLOPROS" models.

## 5.3 Historical Time Series

With the calibrated impact models, we set the flood event of 2022 into historical perspective by computing a time series of monthly flood impacts for the recent years. To that end, we used the daily GloFAS river discharge reanalysis from January 2010 through December 2022 and computed flood footprints from the monthly maximum of the datasets. To represent the uncertainty in our flood model, we employed bootstrap sampling as described in Sec. 3.2 to compute an ensemble of 20 flood footprints for each month. Together with the 20 impact functions from the cross-calibration, we computed $20 \times 20 = 400$ impacts for each month to sample the uncertainty in the impact estimation. Again, we distinguished between one model considering no flood protection and one model considering the FLOPROS protection standards. We chose the matching WorldPop dataset for each year of the time series as exposure (WorldPop, 2020). Because no WorldPop datasets exist for the years 2021 and 2022, we selected the dataset of 2020 for these years, which was also used in the previous calibration step.

The displacement estimates for each model are visualized in Fig. 6, together with timings of flood disasters according to the United Nations Office for the Coordination of Humanitarian Affairs (OCHA, 2023), and with internal displacements due to flood events reported in the Global Internal Displacement Database (IDMC, 2023b). For each month in the time frame considered, the "No Protection" model estimates a median of at least 3,000 people displaced, with an uncertainty of about one order of magnitude. Contrarily, the "FLOPROS" model estimates a median of zero people displaced for most months. For high-impact events like in 2010, 2011, and 2012, the impact estimates of both models are nearly the same. For low-impact events like in 2016 and 2017, the displacement estimates by "FLOPROS" are about one order of magnitude lower than those by the "No Protection" model. The duration of high-impacts during the 2022 floods is similar to the floods of 2015. In terms of estimated displacement, the 2022 floods are comparable to the floods of 2010. For most high-impact events, both models estimate displacement in the same order of magnitude as reported by IDMC (2023b). Except for one instance in January 2017, each flood disaster reported by OCHA (2023) corresponds to a spike in estimated displacement with the same timing.



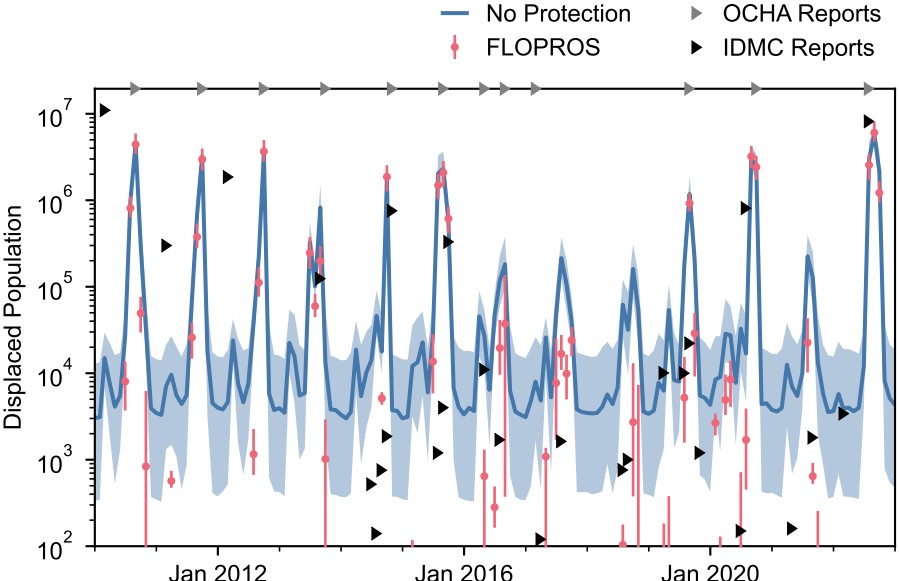

**Figure 6.** Monthly displaced population estimated by the model considering no flood protection and by the model considering FLOPROS protection standards. The solid line ("No Protection") and the markers ("FLOPROS") denote the median of a distribution sampled from 20 bootstrap-sampled flood footprints based on GloFAS river discharge reanalysis for each month, and 20 cross-calibrated impact functions. The shaded areas ("No Protection") and bars ("FLOPROS") depict the range between the 5th and the 95th percentile of the respective distribution. Note that every data point must be interpreted as independent displacement estimate due to river flood occurring in that month. The model considers no previous displacements and thus assumes full recovery after one month. Grey markers at the top axis indicate timings of flood disasters reported by OCHA (2023). Black markers indicate internal displacements due to flood reported by IDMC (2023b). For both types of markers, their left border denotes the beginning of the event as defined by the respective organization.

## 5.4 Impact-based Forecasts

In this section, we apply the flood impact model on the GloFAS river discharge forecast, thus computing an impact-based forecast. The GloFAS river discharge forecast is produced in 24 hour time intervals with a daily time step and a lead time of 30 days, matching the lead time of the ECMWF medium-range weather forecasts which serve as forcing for the GloFAS hydrological model. Like the ECMWF weather forecasts, the GloFAS river discharge forecasts represent model uncertainty by an ensemble of 50 members.

During the 2023 monsoon season, Pakistan has been struck by several flood events, of which one occurred in early July (see ECHO, 2023). We investigated if such an event, and its magnitude with respect to humanitarian impacts, can be predicted by applying the calibrated displacement models to the GloFAS river discharge forecast issued on 08 July 2023. As weather forecasts are typically uncertain for lead times of more than a week, we restricted the forecast window to a lead time of 5 days. For each of the 50 forecast ensemble members, we computed the maximum discharge over the lead time at each location. Then,



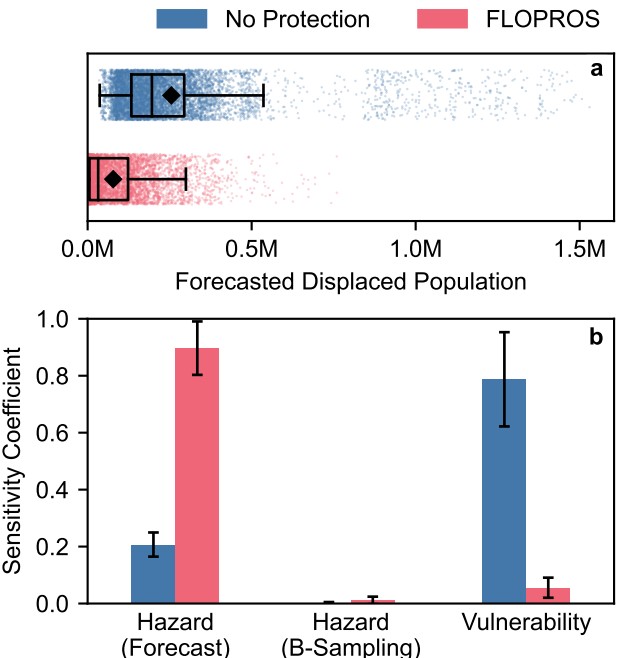

**Figure 7.** Uncertainty and sensitivity analysis of the forecasted displacement in Pakistan based on the GloFAS river discharge forecast from 08 July 2023, including a 5 d lead time, for the two models considering no flood protection and the FLOPROS flood protection standards, respectively. This considers uncertainty in the GloFAS forecast represented by the forecast ensemble ("Forecast"), the statistical flood model uncertainty represented by bootstrap sampling ("B-Sampling"), and the uncertainty in vulnerability represented by the ensemble of calibrated impact functions. **a**: Sampled displacement in Pakistan for both model instances. Each point indicates the total impact of a sample. Boxes denote the lower quartile, median, and upper quartile, and squares the mean. Whiskers delimit the distribution confidence interval, outside which data points can be considered outliers. **b**: First-order sensitivity coefficients with confidence intervals (error bars), indicating the fraction of model output variance that can be attributed to variations in the respective model input.

we employed the flood model and generated 20 flood footprints from each member using bootstrap sampling. We repeated the impact calculation setup of the previous section, with the 20 calibrated impact functions supplying vulnerability information, and the WorldPop dataset of 2020 serving as exposure without uncertainty. For both the model without flood protection and the model considering FLOPROS flood protection standards, this yielded $50 \times 20 \times 20 = 20,000$ impact model combinations. To analyze the uncertainty in the estimated impact and its sensitivity towards the input, we took $2^{13} = 8,192$ samples from each of the two model distributions by employing the uncertainty quantification (unsequa) module in CLIMADA (Kropf et al., 2022).

The estimated displacements for these distributions are displayed in Fig. 7a. The "No Protection" model predicts higher impacts overall, with a median displacement of 195,957 people against the median value of 32,142 for the "FLOPROS" model. The associated means or expected values are 255,211 and 77,959. We find that the impact distributions of both models are





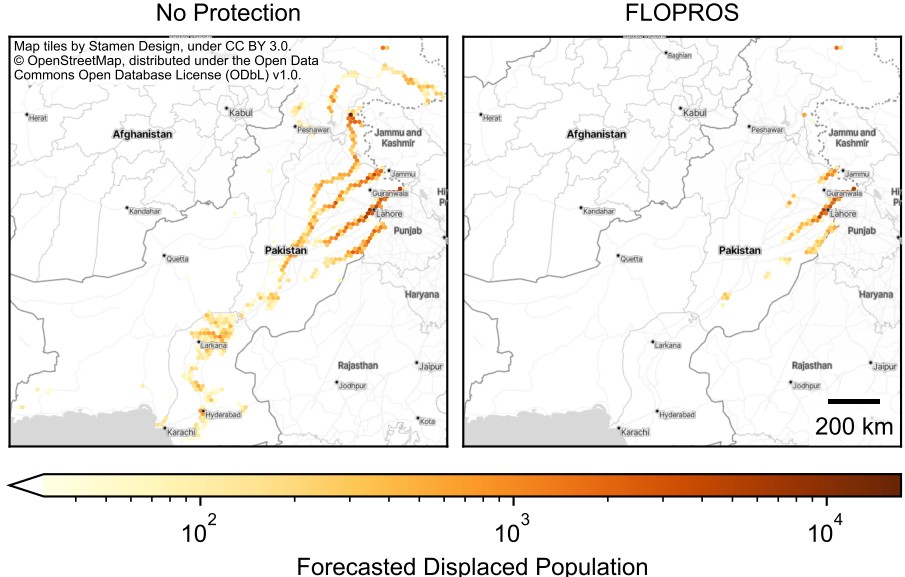

**Figure 8.** Mean forecasted displacement based on the GloFAS river discharge forecast from 08 July 2023, including a 5 d lead time, for the two models considering no flood protection and the FLOPROS flood protection standards, respectively. The color of each hexagon represents the accumulated mean displacement within its area. The impact data itself has the resolution of the WorldPop exposure layer ($30''$). The total expected displacement is 255,211 for "No Protection" and 77,959 for "FLOPROS", see Fig. 7a for the associated distribution.

long-tailed towards higher impacts. The upper limits of the confidence intervals (indicated by whiskers in Fig. 7a) are 536,049 ("No Protection") and 299,525 ("FLOPROS") displaced people, respectively, while outliers exceed 1 million ("No Protection") and 500 thousand ("FLOPROS") displaced people. Both models thus significantly exceed historical displacement estimates for months without reported floods, compare Fig. 6.

The sensitivity of the total estimated displacement $\chi$ towards the input parameter $\pi_i$ can be expressed by the first-order sensitivity coefficient,

$$C_i = \frac{\mathrm{Var}_{\pi_i}\left(\mathrm{E}\left[\chi|\pi_i\right]\right)}{\mathrm{Var}(\chi)} \quad \in [0,1]. \tag{11}$$

where the expectation value in the numerator is evaluated with a fixed parameter $\pi_i$, and the variance in the numerator is evaluated by only changing $\pi_i$ (Saltelli and Annoni, 2010). We interpret each of the input distributions (model forecast ensemble,
bootstrap ensemble, impact function ensemble) as independent input parameter and evaluate their sensitivity coefficients using the aforementioned unsequa module. These coefficients for each model with respective confidence intervals are displayed in Fig. 7b. For the "No Protection" model, the sensitivity coefficient for the GloFAS forecast uncertainty is relatively low with $0.21 \pm 0.04$, and the coefficient for the vulnerability is relatively high with $0.79 \pm 0.17$. This is opposite to the "FLOPROS" model, where the sensitivity coefficient for the forecast uncertainty is high with $0.90 \pm 0.09$ and the coefficient for the vul-
nerability is low with $0.06 \pm 0.04$. Both model estimates feature a negligible sensitivity towards the bootstrap sampling in the





flood model, with sensitivity coefficients of $0.002 \pm 0.004$ ("No Protection") and $0.011 \pm 0.013$ ( "FLOPROS"). See appendix Appendix Sec. B for the uncertainty and sensitivity analysis of forecasts from 06, 07, and 08 July 2023.

Figure 8 displays the spatial distribution of forecasted population displacement. Both the "No Protection" and the "FLOPROS" model estimate high and localized displacement around Lahore in eastern Pakistan. The "No Protection" model addi-
tionally predicts high displacement near the border to Jammu and Kashmir. Further downstream the Indus, around Larkana and Hyderabad, the same model indicates displacement for vast areas, but overall displacement here is lower than in the aforementioned regions. At each location where the "FLOPROS" model predicts displacement, the "No Protection" model does, too. However, the "FLOPROS" estimate is typically lower.

## 6 Discussion

We present a globally consistent flood inundation model integrated and used in an impact model. In the following, we will discuss the results and the performance of the model, as presented in Sec. 5, and relate them to current literature—first in terms of the flood inundation model itself, and then in combination with the impact model.

### 6.1 Flood Inundation Model

The river flood inundation model presented in this paper combines river flood reanalysis and forecast data with river flood
hazard maps, all based on the hydrological modeling system GloFAS. We took care to follow the same statistical approach as Dottori et al. (2016b) for calculating return periods, to ensure that the return periods computed with our model and the return periods of the flood hazard maps are directly comparable. At the time of this writing, GloFAS v4.0 has been operationalized in both the reanalysis and the forecast datasets, featuring major updates to the underlying models and data, and a doubled resolution compared to previous versions (ECMWF, 2023b). The latter, however, implies that return periods of the flood hazard
maps and those computed from GloFAS v4 data are not directly comparable. Switching to GloFAS v4 data in the presented model will therefore require some degree of downscaling and bias correction, akin to methods used in climate modeling. Ideally, GloFAS v4 data should be used together with flood hazards maps computed from the same model. The model could likewise be applied using river discharge data from the European Flood Awareness System (EFAS; Thielen et al., 2009) and flood hazard maps for Europe by Dottori et al. (2022) to achieve a higher accuracy and footprint resolution in Europe (only).
Contrary to Alfieri et al. (2015), who assigned areas of flood risk to the grid point of the corresponding hydrograph, the presented flood model operates on a local basis. For each location within a flood plain, the return period and flood inundation are computed separately, considering neither neighboring grid cells (except for the interpolation during regridding) nor the discharge inside the corresponding river. This has two major implications. On the one hand, neither Van Der Knijff et al. (2010) nor Alfieri et al. (2013) mention that the process of flooding, where water leaves the routing channel of the river, is
explicitly considered in the hydrological LISFLOOD model used in GloFAS. We therefore have to assume that flooding in very large river plains might be underestimated in our model because local discharge might not adequately reflect that water is leaving the river bed and spilling into the flood plain. However, this effect is slightly counteracted by the difference in resolution





between the hazard maps and the discharge data, as the latter represents each river with a width of 0.1 degrees latitude and longitude. On the other hand, the model is able to represent pluvial floods in river basins to some degree. In situations where
discharge around a river is unusually high, but the river is able to take up that discharge without overflowing, the flood model will estimate a flooded area that would not have been predicted when only taking into account the return period at the location of the river. Such a situation is displayed in Fig. 1 towards the north-west, where the return periods around the river are much higher than for the discharge in the river itself. Nonetheless, fluvial and pluvial flood models are clearly distinct, and our model should not be considered as the latter (cf. Eilander et al., 2023).

We compared the performance of the model with and without FLOPROS flood protection levels included through binary classification metrics displayed in Table 1. According to these metrics, and specifically for the case of the Pakistan 2022 floods, including FLOPROS protection levels does not clearly improve the estimated flood extent. With FLOPROS, the specificity of the model increases strongly, and the precision slightly. However, the recall clearly decreases. While including FLOPROS thus avoids false positives considerably, it also reduced the true positive rate of the model. This is in line with recent studies, which
found that FLOPROS tends to overestimate flood protection levels, causing flood models to underestimate flood extents and severity (e.g. Mester et al., 2021). Furthermore, the classifiers should not be interpreted too thoroughly, because the degree to which flood models can be verified is inherently limited, especially when relying on satellite observations (Bates, 2023).

## 6.2 Impact Model

The impact model combines the flood inundation model with the CLIMADA impact framework. The inundation model there
supplies the hazard component. The vulnerability is determined by calibrating impact functions to impact data of past events.

Although the flood model shows significant differences to the satellite-observed flood extent, the impact functions can be reasonably calibrated to replicate the number of displaced people in Sindh, Pakistan during the 2022 floods. For nearly all districts, the number of displaced people estimated by the calibrated model deviates less than order of magnitude from the reported number. We chose a step function as vulnerability for simplicity. The cross-calibrated parameters spread considerably,
see Fig. 4, but the effect of this spread on the estimated impact is surprisingly low, see Fig. 5. Likewise, although the distribution of parameters is different, the model with FLOPROS protections standards estimates impacts very similar to the model without flood protection measures. However, this might also be an effect of only slight differences between both footprints. We can therefore conclude that the deviation between estimated and reported impact is due to systematic model errors, like errors in the flood footprint or due to the particular choice of impact function.

Because of the highly simplified impact function, interpreting the calibrated impact function parameters is difficult. It is natural to assume that displacement increases with increasing flood height. Since this is not covered by the impact function, the displaced population ratio of around 25 % must be interpreted as the statistical mean over all occurrences of displacement, regardless of the local flood level. Still, the calibrated threshold parameter of around 0.5 m appears suitable to indicate a severe flood that might cause displacement. For instance, Kam et al. (2021) assumed a lower inundation threshold of 0.5 m for
estimating displacement risk due to river floods. Overall, the calibrated parameters follow the expected behavior; to match the calibration data, the model with the larger flood footprint ("No Protection") must estimate a lower impact per location, hence





the median impact threshold is higher and the median percentage of displaced population is lower than for the model with smaller flood footprint ("FLOPROS").

Fig. 5 further indicates that the flood impact model underestimates displacement in rural districts like Umer Kot and Ja-
cobabad, and overestimates displacement in densely populated districts like Hyderabad and Karachi. This likely relates to the effects of the flood leading to displacement, and our simplifying model assumptions. In the model, displacement can only occur if inhabited areas are flooded. In rural areas, however, people might be displaced due to the lack of food, disruption of infrastructure, or destruction of farmland, even if their homes are technically unaffected by the flood. The densely populated larger cities, on the other hand, might have better flood protection and disaster mitigation measures. Indeed, large areas of
farmland were destroyed and a considerable number of livestock perished as a result of the flood (OCHA, 2022a), and larger cities like Karachi received thousands of people from flood-affected rural areas (Tunio, 2022).

The historical time series of displacement estimates in Pakistan displayed in Fig. 6 reveals that the "No Protection" model estimates a "baseline" of 1,000 to 10,000 displaced people each month. While these figures are clearly exaggerated, they are an effect of the model assuming no flood protection at all. Nonetheless, high-impact flood events can be clearly distinguished
from that baseline. The "FLOPROS" model predicts no displacement for most months. Therefore, any impact estimate above zero by this model is indicative of a flood event.

The timings of events estimated by the models match the reported data well. For high-impact events, the order of magnitude in displaced population matches the reported numbers for both models. However, since these events often span several months, the exact numbers are difficult to compare because in our particular model setup, we calculated impacts for each month
separately, thus assuming full recovery after one month. Since 2014, IDMC reports include many lower-impact flood events. While not all of them coincide with a peak in displacement estimated by the models, most of them range within or below the "No Protection" model baseline. The "FLOPROS" model therefore seems better suited for identifying flood disasters, and estimating their severity.

With these results, we assume that the model can also be applied for identifying high-impact disasters if no data for cali-
bration is available. If the parameters of the impact function are chosen such that the impact estimates become sensitive to the input, high-impact events will become evident by a peak in estimated displacement that can be clearly distinguished from the model baseline. Our calibration results of the displacement data in Pakistan indicate that a flood inundation threshold of about 0.5 m might be a suitable measure. The event severity can then only be described relative to past events.

As depicted in Fig. 8, both models can provide spatial information on flood impact hotspots. Comparing their output may
provide information on worst-case scenarios, as the "No Protection" model effectively models the expected impact in case all protection measures fail. As we established that the "FLOPROS" model flood footprint is not necessarily a better estimate for the true flood extent, the "No Protection" model should therefore not be discarded.

Applying the impact model to a GloFAS river discharge forecast again revealed significant differences between the "No Protection" and the "FLOPROS" model. The exact sensitivity of the impact estimate varies between forecasts. As shown in
Fig. 7b, the "FLOPROS" model is more sensitive to the variation in the hazard forecast, and less sensitive to the vulnerability than the "No Protection" model. However, these findings are unique to the forecast from 08 July 2023 and the particular





model setup, and need not follow a general trend. Nonetheless, there is evidence that the sensitivity towards input parameter uncertainty is stable, at least for a single flood event, see Appendix Sec. B. The differences in sensitivity between the two models can be explained with the application of the flood protection and the calibrated impact functions. For the "FLOPROS"

model, the most important information is whether the protection level is exceeded or not. If it is, the flood depth is typically above the thresholds of the impact function ensemble. Therefore, sensitivity towards the hazard is higher than towards the vulnerability. The "No Protection" model is much more sensitive towards vulnerability because flooding occurs inevitably and the vulnerability therefore controls most of impact magnitude. For both models, the sensitivity towards bootstrap sampling is negligible against the forecast uncertainty.

## 7 Conclusions

We presented a model for mapping river flood inundation footprints to GloFAS river discharge data. Its major advantage compared to physical river routing and flood dynamics models is the relatively low computational effort required to generate these footprints. The model is globally applicable, harnessing the high data quality of the GloFAS products. It is readily implemented in the natural catastrophe impact modeling framework CLIMADA. We applied this model to estimate population

displacement due to river floods in Pakistan, calibrating an ensemble of impact functions based on displacement data from 2022. We then applied the model to estimate a historical time series of displacements due to floods in Pakistan that matches OCHA and IDMC disaster reports well. We further demonstrated that this flood impact model can be applied to detect imminent events and estimate flood disaster severity through impact-based forecasts. While we showed that the model performs well in terms of country-wide numbers, we found significant differences on the district level between the calibrated model impact and

reported displacements. We therefore conclude that the model's strengths lie in estimating overall event impacts, and identifying spatial hotspots, rather than small-scale flood dynamics analysis, e.g. on the city level. Although incorporating estimates of flood protection standards from the FLOPROS database changes flood footprints significantly, its effects on overall model performance remain inconclusive. We suggest to use both model versions to estimate "worst-case" and "best-case" scenarios, but a detailed comparison of the estimated impacts warrants further research. A sensitivity analysis revealed that the statistical

uncertainty within our model is negligible compared to the uncertainty represented in the GloFAS river discharge forecast and the cross-calibrated impact functions. However, this analysis did not consider systematic model errors, such as uncertainty in the flood hazard maps or the exposure layer in the impact model. Still, dissecting the overall uncertainty in the estimated impact into sensitivity coefficients for each input parameter provides crucial information for decision makers, as major sources of uncertainty can be identified. Further work on this flood model and its overall approach should focus on operationalizing

early event detection and classification, thus supporting humanitarian organizations and stakeholders in anticipatory action and decision making.



## Appendix A: Classification Metrics for Flood Extent

The metrics for binary classification, indicating the predictive performance of the flood extent estimated by the model against the flooded area observed by satellites are computed as follows.

With true positive (TP) we denote the intersection of the flooded area estimated by the model $A_m$ with the flooded area observed by the satellite $A_s$,

$$A_{\mathrm{TP}} = A_m \cap A_s. \tag{A1}$$

Accordingly, a false positive (FP) is the set difference between $A_m$ and $A_s$, and vice versa for a false negative (FN),

$$A_{\mathrm{FP}} = A_m \setminus A_s, \quad A_{\mathrm{FN}} = A_s \setminus A_m. \tag{A2}$$

The classification measures precision $P$ and recall $R$ are calculated from these areas according to

$$P = \frac{\|A_{\mathrm{TP}}\|}{\|A_{\mathrm{TP}}\| + \|A_{\mathrm{FP}}\|}, \tag{A3}$$

$$R = \frac{\|A_{\mathrm{TP}}\|}{\|A_{\mathrm{TP}}\| + \|A_{\mathrm{FN}}\|}, \tag{A4}$$

where $\|\cdot\|$ indicates the total area computed in cylindrical equal area projection (ESRI:54034). We denote the set difference between the area of the most extensive river flood hazard map $A_h$ (for a 500-year return period) and the union of observed and modeled flood areas as the true negative (TN),

$$A_{\mathrm{TN}} = A_h \setminus [A_m \cup A_s]. \tag{A5}$$

We chose $A_h$ instead of the whole area of Pakistan because the river flood model can only estimate flooding in the area of the flood plains represented in the hazard maps. Choosing the whole area of Pakistan that was not flooded would artificially improve the model classification score. With this, we can compute the specificity as

$$S = \frac{\|A_{\mathrm{TN}}\|}{\|A_{\mathrm{TN}}\| + \|A_{\mathrm{FP}}\|}. \tag{A6}$$

The critical success index (CSI) compares true positive against false negative and false positive,

$$\mathrm{CSI} = \frac{\|A_{\mathrm{TP}}\|}{\|A_{\mathrm{TP}}\| + \|A_{\mathrm{FN}}\| + \|A_{\mathrm{FP}}\|}, \tag{A7}$$

and the F1-score is given by the harmonic mean between precision and recall,

$$F_1 = 2\frac{PR}{P+R}, \tag{A8}$$

These measures take values between 0 and 1, with 1 indicating perfect prediction. Finally, the Matthews Correlation Coefficient (MCC) computes the correlation between predicted and measured values as

$$\mathrm{MCC} = \frac{\|A_{\mathrm{TP}}\|\,\|A_{\mathrm{TN}}\| + \|A_{\mathrm{FP}}\|\,\|A_{\mathrm{FN}}\|}{\sqrt{[\|A_{\mathrm{TP}}\| + \|A_{\mathrm{FP}}\|]\,[\|A_{\mathrm{TP}}\| + \|A_{\mathrm{FN}}\|]\,[\|A_{\mathrm{TN}}\| + \|A_{\mathrm{FP}}\|]\,[\|A_{\mathrm{TN}}\| + \|A_{\mathrm{FN}}\|]}} \quad \in [-1, 1], \tag{A9}$$

where 1 indicates perfect agreement, 0 indicates the predictive capability of random values, and -1 indicates complete disagreement.





**Table B1.** Forecasted displacement by the models including no protection standards ("No Protection") and "FLOPROS" protection standards based on GloFAS river discharge forecasts issued on 06, 07, and 08 July 2023. The reported values are means and medians of the impact distributions yielded by both models, representing uncertainty from the discharge forecast, the bootstrap sampling within the flood model, and the ensemble of calibrated impact functions, see Sec. 5.4.

|              | No Protection |         | FLOPROS |         |
|--------------|---------------|---------|---------|---------|
|              | Mean          | Median  | Mean    | Median  |
| 06 July 2023 | 379,511       | 204,749 | 133,177 | 6,250   |
| 07 July 2023 | 249,522       | 159,066 | 95,659  | 22,948  |
| 08 July 2023 | 255,211       | 195,956 | 77,958  | 32,141  |

## Appendix B: Uncertainty and Sensitivity Analysis of Multiple Forecasts

We expect that the results of the uncertainty and sensitivity analysis of the forecast as given in Sec. 5.4 do not necessarily follow a general trend. It is conceivable that the sensitivity towards bootstrap sampling becomes larger when the uncertainty in the river discharge forecast is very low. Likewise, the interplay of FLOPROS protection standard and possible inundation depths might be different in other countries, increasing the sensitivity towards vulnerability even for the "FLOPROS" model. Additionally, the uncertainty of the river discharge forecast, and hence the associated sensitivity, strongly depend on the chosen lead time. Finally, the spread of estimated impacts can be much lower if the model does not predict a significant flood for a given river discharge forecast.

We display the model uncertainty of forecasted displacement based on GloFAS forecasts issued on 06, 07, and 08 July 2023, including a 5 day lead time each, in Fig. B1. The associated distribution means and medians are given in Table B1. While mean and median for the "No Protection" model stay relatively stable and are within the same order of magnitude, the "FLOPROS" model exhibits a much more skewed distribution. The days before 08 July 2023, mean and median differ significantly, and the median increases from 6,250 to 32,141 displaced people over two days. At the same time, the distribution spread of the "No Protection" model decreases from around 1 million to 500 thousand displacements as the upper confidence interval limit.

The results of the sensitivity analysis of the same forecasts are displayed in Fig. B2. While the sensitivity coefficients for the "FLOPROS" model show the same pattern as in Fig. 7 for all forecasted days, the sensitivity of the "No Protection" model output shifts from forecast to vulnerability. We surmise that this is due to a nearing flood event for which the forecast uncertainty reduces the more imminent it becomes. This reduces the spread of the overall forecasted displacement. At the same time, it becomes more and more apparent within the forecast, that protection levels as given in the FLOPROS database might be exceeded, demonstrated by a clear increase in median displacement in the "FLOPROS" model. This also contributes to the slightly increased sensitivity towards vulnerability in that model.

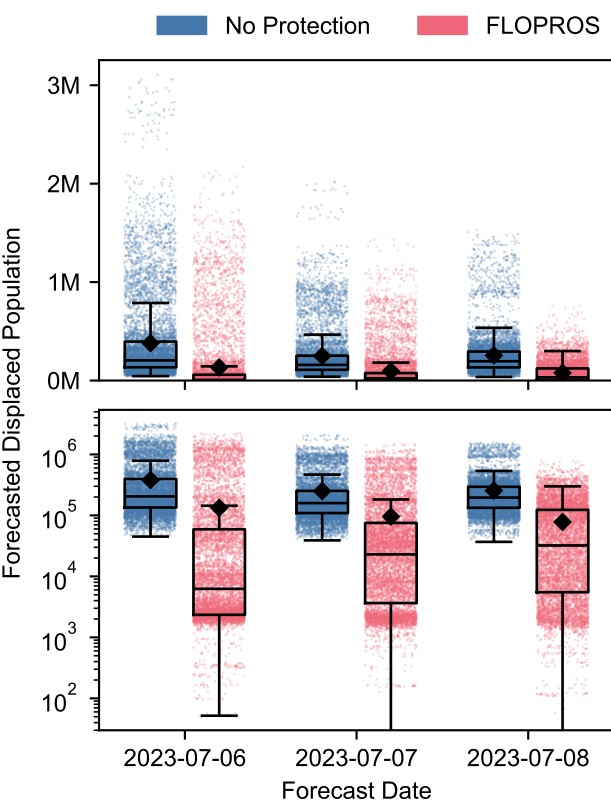

**Figure B1.** Sampled displacement in Pakistan for the forecasts from the model considering no flood protection ("No Protection") and the model considering "FLOPROS" flood protection standards. Each point indicates the total impact of a sample. The sampling considers uncertainty represented by the GloFAS discharge forecast ensemble, by the bootstrapping when calculating the return periods within our model, and by the impact function ensemble calibrated with the respective model. Boxes denote the lower quartile, median, and upper quartile, and squares the mean. Whiskers delimit the distribution confidence interval, outside which data points can be considered outliers. The subplots display the exact same data in linear (**top**) and in logarithmic (**bottom**) scale.



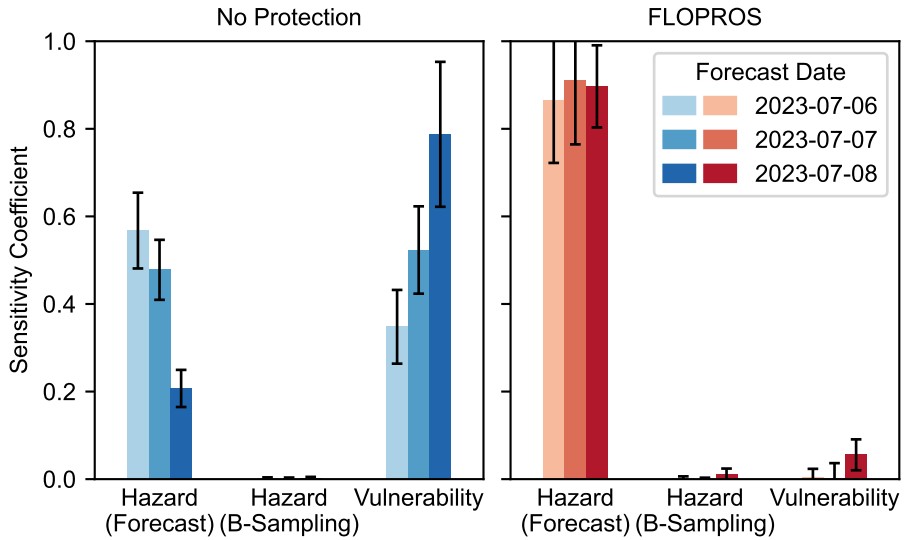

**Figure B2.** First-order sensitivity coefficients with confidence intervals (error bars) for the impacts estimated by the model considering no flood protection ("No Protection") and the model considering "FLOPROS" flood protection levels, based on GloFAS forecasts issued on 06, 07, and 08 July 2023, including a 5 day lead time each. "Forecast" denotes the uncertainty represented by the GloFAS discharge forecast ensemble, "B-Sampling" the uncertainty from bootstrap sampling when computing the return period within our model, and "Vulnerability" the uncertainty represented by the impact function ensemble calibrated with the respective model. The sensitivity coefficients can be interpreted as indicating the fraction of impact variance that can be attributed to variations in the respective model input.



*Author contributions.* LR and TR conceptualized the model and its application. LR developed the software, retrieved the data, conducted the research, created the visualizations, and wrote the manuscript draft. TV reviewed the software. All authors reviewed and edited the manuscript, and participated in continuous discussions.

*Competing interests.* The authors declare no competing interests.

*Code and data availability.* Software, data, and scripts for replicating the computations within this publication are available from https://doi.org/10.5281/zenodo.10518953 (Riedel, 2024).

*Acknowledgements.* The authors thank Chahan M. Kropf (ETH Zürich), Isabelle Bey (MeteoSwiss), and Pamela Probst (MeteoSwiss) for their valuable comments on the paper draft.



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
