# Peer review of "Fluvial Flood Inundation and Socio-Economic Impact Model Based on Open Data"

_EGUsphere, 2024_

## Author Comment (AC1)

**Fluvial Flood Inundation and Humanitarian Impact Model Based On Open Data**

**Authors' Response**
**To Referee Comment (RC) 1**

Lukas Riedel on behalf of the authors

12 April 2024

**1 Comments**

**Referee Comment (RC)**  Firstly, I want to express my gratitude for giving me the opportunity to review this excellent piece of work. I must admit that as I am not a risk modeler, I may not be able to provide detailed commentary on the models integrated into this paper. Overall, I found the paper to be well-structured and thoughtfully presented. However, I do have a few suggestions for modifications that I believe could enhance clarity and readability. These suggestions are outlined below.

**Author Response (AR)**  Thank you for a positive and thorough review. We appreciate the added interdisciplinary perspective and we feel confident that it strongly improved the paper. For additional context, see our detailed responses below.

**RC**  In the introduction but more importantly in the title, the word "humanitarian" in this context might be misleading. While floods can indeed have significant humanitarian impacts, the term "humanitarian" typically refers to actions or interventions aimed at alleviating human suffering, particularly in emergency situations. I will highly recommend revising the introduction to better convey the challenges faced by communities affected by floods.

**AR**  Thank you for this correction. It was difficult to find an umbrella term for the impacts our model could consider. We wanted to clearly distinguish the socio-economic impact model from the physical flood inundation model, and to indicate that the model can be calibrated to other types of impacts as well (if suitable data is available). After reconsideration, we decided to rename the manuscript title to "Fluvial Flood Inundation and Socio-Economic Impact Model Based on Open Data".

**RC**   The second paragraph of the introduction refers to the Sendai Framework for Disaster Risk Reduction. The correct footnote should cite UNDRR instead of UNISDR.

**AR**   We corrected the reference to (UNDRR, 2015).

**RC**   It would also be beneficial to define displacement in the introduction, particularly in the context of flooding, such as during monsoon seasons. This is important because individuals may experience multiple waves of flooding, resulting in repeated displacement.

**AR**   We added a definition of disaster displacement to the introduction.

**RC**   I would suggest merging sections 2) "Data" and 3) "Flood Model" into one section. The rationale behind this suggestion is that the data section currently contains a significant amount of information on various flood modeling techniques. I recommend using the data section as an introductory section instead. Later in the document, this data could then be utilized to present empirical evidence from past events from multiple sources.

**AR**   We merged the "Data" section (previously sec. 2) into the "Flood Model" section (sec. 2), renaming it to "Input Data" (sec. 2.1).

**RC**   Section 3 flood model: I would suggest revising the 1st sentence:
"To compute a flood inundation footprint from gridded, geo-located river discharge data, said data is related to the historical 85 discharge time series via an extreme value analysis, and the corresponding return period is used to look up flood depths in flood hazard maps."

**AR**   This sentence was indeed hard to grasp. We simplified the introduction to "Flood Model" (sec. 2, previously sec. 3).

**RC**   In Section 4, Implementation, I suggest refraining from using the term 'natural,' especially when discussing exposure. You can explore the 'no natural disasters' campaign, which emphasizes that while some hazards are inherent in nature and unavoidable, the resulting disasters are often influenced by human actions and decisions. `https://www.nonaturaldisasters.com/`

**AR**   Thank you for raising awareness of this topic. We concur and removed any mention of "natural disaster" or "natural catastrophe" throughout the document.

**RC**   Regarding section 5.3 on historical times series. Have you reach out to the Pakistan government to explore further historical trends? I think data are available only since 2017 `https://pdma.punjab.gov.pk/`

**AR**   The data provided in the situational reports by PDMA Sindh and Punjab are typically in PDF format, and sometimes only in image data formats. These formats make it difficult to machine-read the data, which heavily complicates parsing larger amounts of data to establish historical trends. We reached out to PDMA Sindh to possibly receive machine-readable tabular data of their situational reports, but have not received a response. The data provided by the IDMC and OCHA databases therefore were our only sources of historical trends.

**RC**   I highly recommend adding a section on Vulnerability to explain how it is defined for flood displacement as an impact function. We can find some information in section 6.2 where you mention that vulnerability is determined by calibrating impact functions to impact data from past events. It would be beneficial to expand on this further. How many events were assessed? What were the dates and magnitudes? Additionally, the choice of the step function for simplicity may require additional information.

**AR**   We expanded the explanation of the vulnerability in a new subsection "Impact Model and Calibration Setup" (sec. 4.2.1), and added a table listing the extracted PDMA displacement data in Appendix B. For means of the calibration, the displacement during the 2022 floods was considered a single event, whose flood footprint was the maximum flood extent and inundation between 01 July and 30 September 2022.

**RC**   Later, you mention in your calibration process that you decided to use 0.5m as a threshold for Pakistan. Having this information centralized in one section would make it easier to understand. It appears that you calibrated to 0.5m for Pakistan. Do you have any measurements that can corroborate this hypothesis? Additionally, it could be interesting to explain in the conclusion how, as you aim to develop a global impact forecasting system, parameters may be adjusted depending on the context.

**AR**   We updated the impact model discussion (sec. 5.2, previously sec. 6.2) to clarify that we do not set a threshold of 0.5m ourselves, but that Kam et al. (2021) chose a threshold of 0.5m for calculating displacement risk due to river floods in a pessimistic scenario. We further clarify that the medians of the inundation thresholds calibrated with our model (No Protection: 0.58m. FLOPROS: 0.52m) are close to this value. To make the distinction clearer, we also revised the paragraph on using the model in situations where no calibration data is available. We now exemplify using a threshold of 0.2m to indicate population at risk of flooding, to further differentiate from our previous model setup.

**RC**   At around line 390, I would recommend changing the wording from "historical time series of displacement" to "historical trends". The reason for this change is that each displacement has a duration that can vary from hours to days, months, or years. The term "time series" typically refers to situations where we have multiple snapshots of information over time. Regarding

the flooding events of 2022, there are still more than 1.1 million people living in displacement situations in Sindh province. They were not able to return home due to many obstacles.

**AR**   Indeed, it was not clear enough what the model reports. In terms of the Global Internal Displacment Database (IDMC, 2023), it reports "Internal Displacements" for each month, and not "Internally displaced people (IDPs)" over time. It also assumes full recovery after each month.
We removed any mention of "historical time series" and instead refer to "monthly displaced population" and "historical (flood) displacement". We changed the subsection title from "Historical Time Series" (previously sec. 5.3) to "Historical Flood Displacement" (sec. 4.3). We further added the caveat on the model assumption of full recovery after one month to this section and placed it more prominently in the caption of fig. 6.

**RC**   In the conclusion I personally welcome the suggestion of a range of people at risk of displacement between the worst case and best-case scenarios.

**AR**   Unfortunately, we cannot state a general range of people at risk in best- and worst-case scenarios. We slightly updated the conclusion, trying to further clarify this issue. Referring to "Impact-based Forecasts" (sec. 4.4, previously sec. 5.4) and fig. 7a in particular, we find that estimated impacts between the two models ("No Protection", "FLOPROS") may differ a lot. But the uncertainty within each model is also considerable. Recommending a specific workflow to determine reasonable best- and worst-case scenarios indeed warrants further research. The uncertainty and sensitivity analysis in sec. 4.4 (previously sec. 5.4) exemplifies this.

**2  References**

IDMC: Global Internal Displacement Database, Internal Displacement Monitoring Centre (IDMC), URL `https://www.internal-displacement.org/database/`, 2023.

Kam, P. M., Aznar-Siguan, G., Schewe, J., Milano, L., Ginnetti, J., Willner, S., McCaughey, J. W., and Bresch, D. N.: Global warming and population change both heighten future risk of human displacement due to river floods, Environmental Research Letters, 16, 044 026, https://doi.org/10.1088/1748-9326/abd26c, 2021.

UNDRR: Sendai Framework for Disaster Risk Reduction 2015 - 2030, United Nations Office for Disaster Risk Reduction (UNDRR), URL `https://www.undrr.org/quick/11409`, 2015.

---

## Author Comment (AC2)

**Fluvial Flood Inundation and Humanitarian Impact Model Based On Open Data**

**Authors' Response**

**To Referee Comment (RC) 2**

Lukas Riedel on behalf of the authors

12 April 2024

**General Comments**

**Referee Comment (RC)**  The paper presents a valuable contribution to the field of fluvial flood modeling and forecasting, addressing a critical need for efficient and accurate methods to assess flood impacts globally. The use of extreme value analysis coupled with openly available data within a catastrophe modeling framework offers a promising approach to rapidly compute flood inundation footprints and estimate associated impacts. The application of the model in Pakistan exemplifies its utility in assessing flood depths, extents, and population displacement. The findings regarding the incorporation of estimated flood protection standards and the calibration of vulnerability models provide important insights for future research and disaster preparedness efforts.

**Author Response (AR)**  Thank you for an encouraging and concise review. We are happy to see that we agree on many positive aspects of the presented research. By addressing the raised concerns, we were able to sharpen aspects of the discussion in the manuscript (sec. 5, previously sec. 6), and we provide additional context in our detailed responses below.

**RC**  I would be excited to see how this model could translate into better impact-based early warning systems, potentially revolutionizing the way we prepare for and respond to fluvial flood events. Additionally, while the methodology is tested in Pakistan, it would be beneficial to explore how it could be extended to other countries and contexts, considering varying environmental and socioeconomic factors.

The paper addresses the main sources of uncertainty, including uncertainty in displacement data, river discharge, and flood footprint, in a convincing manner. However, it is acknowledged that these factors remain significant limiting factors that could influence the accuracy and reliability

of the model's predictions. Further research and improvements in handling uncertainty will be crucial for enhancing the robustness and applicability of the model on a global scale.

**AR**   Indeed, while we consider the model to be globally applicable, it is beyond the scope of this study to ascertain its robustness and accuracy on a global scale. In this publication we want to present the methodology and exemplify the application of the model. In future research, we expect to apply this model in different regions of the globe, in different time frames and scales, and with qualitatively different data. This will necessarily involve fine-tuning to case-specific environmental and socio-economic factors, which might eventually lead to the development of a global-scale impact model methodology.

In a project by the Federal Office of Meteorology and Climatology MeteoSwiss, we are working closely with stakeholders at the World Meteorological Organization (WMO) and humanitarian organizations to prototype impact estimation and decision making support based on this model and other hydro-meteorological impact models.

**Specific Comments**

**RC**   Line 228: maximum displaced population is reported a month after the end of the peak season. Is this due to a delay in reporting or because of multiple waves in displacements. Typically, we see an initial wave for people directly impacted and a 'late' displacement wave due to other socioeconomic impacts such as loss of livelihoods, services, market disruptions etc. The latter is not expected to be captured by the model.

**AR**   This touches an important aspect of displacement that is difficult to incorporate into our model setup. Technically, the model only considers displacement due to flooding of residential buildings or areas. However, the vulnerability in the model is calibrated using displacement data from IDMC, which also includes displacement due to the mentioned "late" or "secondary" effects. The model thus cannot attribute displacement to specific reasons. Nonetheless, the overall numbers including displacement from direct and indirect impacts should be comparable, if there is little spatial disparity between primary and secondary drivers of displacement. We discussed this in part in Section 5.2 (previously 6.2), but have expanded the respective paragraph and the model setup subsection (sec. 4.2.1) to better reflect these considerations.

Since the model uses the flood footprint as hazard, expanding the model to include the temporal dimension of displacement is not easy. The model will report the overall displacement expected from the flood footprint superimposed on the population distribution, irrespective of when this displacement actually occurs. The calibration only yields sensible results if larger flooded areas and greater inundation depths lead to more displacement. Secondary displacement waves, that occur when the actual flood water has receded, cannot be captured with this model setup. This is why we calibrated the model with the maximum flood footprint of an arguably large time span (June through September 2022) to the reported displacement on 30 Sept 2022.

**RC**   Line 369: It is indeed surprising that such a large variation in the impact functions result in a relatively small difference in displacement. I am wondering if this could be due to some sort of overfitting of the data as we have limited displacement data and a pretty wide parameter space that is considered?

**AR**   Overfitting can be a major issue when calibrating to a single event, and we tried to avoid it by employing the "cross-calibration" approach. We interpret the small difference in displacement for varying calibrated impact functions as an indication for a "stable" calibration; We can, at random, exclude data points from the calibration and still receive similar results. The results of section 4.3 (previously 5.3) "Historical Flood Displacement" indicate that the calibrated impact functions are indeed transferable to displacement events of the last decade in Pakistan. The seemingly high ambiguity of impact function parameters might be due to our particular choice of a step function. We intend to investigate the calibration of other impact function shapes in the future. Nonetheless, we can interpret the calibrated impact function parameters further. For the "No Protection" model, we find that a greater flood inundation threshold $T$ coincides with a greater ratio of displaced population $\Pi$ for the calibrated function (cf Fig. 4). Therefore, although the function parameters have a relatively high spread, the overall displacements resulting from these functions can be similar—note that a greater $T$ decreases the impact while a greater $\Pi$ increases it. For the "FLOPROS" model, we do not find such a clear relationship. However, this model effectively applies two thresholds; first the protection standard threshold is applied to the hazard footprint, and then the flood inundation threshold is applied through the impact function. If the FLOPROS threshold relates to a similar or greater inundation depth than the impact function threshold parameter, it is expected that the effect that parameter is reduced. Therefore, the sensitivity towards the parameter is low, and hence its spread is large (compare the reduced vulnerability sensitivity coefficient for "FLOPROS" in fig. 7). Due to the threshold of the flood protection standard, this again need not result in a large spread in displacement figures.